



# A 60–year (1961–2020) near-surface air temperature dataset over the glaciers of the Tibetan Plateau

Jun Qin[1,*], Weihao Pan[2,3], Min He[1,2], Ning Lu[1], Ling Yao[1], Hou Jiang[1,*], Chenghu Zhou[1]

[1]State Key Laboratory of Resources and Environmental Information System, Institute of Geographic Sciences and Natural Resources Research, Chinese Academy of Sciences, Beijing, China.
[2]University of Chinese Academy of Sciences, Beijing 100049, China.
[3]Key Laboratory of Tibetan Environmental Changes and Land Surfaces Processes, Institute of Tibetan Plateau Research, Chinese Academy of Sciences, Beijing 100101, China.

*Correspondence to*: Jun Qin (qinjun@igsnrr.ac.cn);  Hou Jiang (jiangh.18b@igsnrr.ac.cn)

**Abstract.** Surface air temperature (SAT) is a key indicator of global warming and plays an important role in glacier melting. On the Tibetan Plateau (TP), there exist a large number of glaciers. However, station SAT observations on these glaciers are extremely scarce, and moreover the available ones are characterized by short time series, which substantively hinder our deep understanding of glacier dynamics due to climate changes on the TP. In this study, an ensemble learning model is constructed and trained to estimate glacial SATs with a spatial resolution of $1\,\text{km} \times 1\,\text{km}$ from 2002 to 2020 using monthly MODIS land surface temperature products and many auxiliary variables, such as vegetation index, satellite overpass time and air pressure. The satellite-estimated glacial SATs are validated against SAT observations at glacier validation stations. Then, long-term (1961–2020) glacial SATs on the TP are reconstructed by temporally extending the satellite SAT estimates through Bayesian linear regression. The long-term glacial SAT estimates are validated with root mean squared error, mean bias error, and determination coefficient being 1.61 ℃, 0.21 ℃, and 0.93, respectively. The comparisons are conducted with other satellite SAT estimates and ERA5-Land reanalysis data over the validation glaciers, showing that the accuracy of our satellite glacial SATs and their temporal extensions are both higher. The preliminary analysis illustrates that the glaciers on the TP as a whole have been undergoing a fast warming but the warming exhibits a great spatial heterogeneity. Our dataset can contribute to the monitoring of glaciers' warming, analysis of their evolution, etc. on the TP. The dataset is freely available from the National Tibetan Plateau Data Centre at https://doi.org/10.11888/Atmos.tpdc.272550 (Qin, 2022).

## 1 Introduction

Surface air temperature (SAT) represents the thermal state of the lower atmosphere and serves to regulate the earth surface energy and water budgets and thus impacts the land-atmosphere interaction (Pratap et al., 2019; Huang et al., 2019) and is one of the most important variables in ecology, hydrology, climatology, and environmental sciences (Rasouli et al., 2022; Trebicki, 2020; Box et al., 2019; Huang et al., 2019; Tong et al., 2018). SAT, as a key indicator of global climate change,



has been rising rapidly since the industrial revolution, inducing global glacier mass losses (Radić et al., 2014; Hock et al., 2019). The Tibetan plateau (TP) is the highest plateau and designated as the roof the world and the third pole (Yao et al., 2019). At the same time, the TP owns the maximum number of glaciers, harboring the largest solid water reserves in the world apart from the polar regions, and thus named as the Asian water tower (Pfeffer et al., 2014; Rounce et al., 2020a; Yao
et al., 2012). Similarly, the TP has also been undergoing such a warming (Peng et al., 2021) and even exhibits a more intense warming than its surrounding areas (Nie et al., 2021; Lalande et al., 2021; Bhattacharya et al., 2021). This accelerates glaciers to melt and retreat on the TP (Pratap et al., 2019; Rounce et al., 2020a; Brun et al., 2017; Bhattacharya et al., 2021; Shean et al., 2020; Farinotti et al., 2020), inducing a series of glacial catastrophes, such as glacier lake outburst floods and threatening the neighboring community and infrastructure (Miles et al., 2021; Immerzeel et al., 2020; Kraaijenbrink et al.,
2017). So, an extensive monitoring of glacial SATs on the TP is significant for deeply understanding the response of glaciers to climate change and securing the local residents from possible glacial hazards (Guo et al., 2019).

However, the knowledge of the spatial patterns of glacial SATs and their changes on the TP have still been full of great uncertainties due to the lack of observations (Rounce et al., 2020b). Almost no station SAT observations were available on the glaciers of the TP in the past owing to both poor logistics and limited funding (Qin et al., 2009). Many scientists recently
have tried to deploy automatic weather stations on the glaciers of the TP to collect meteorological variables including SAT and publicized these field observations (Wei, 2021; Yinsheng, 2018b, a; Huabiao, 2021). On the other hand, it is almost impossible to regularly maintain these instruments due to harsh natural conditions on the glaciers. Thus, the duration of glacial SAT observations is usually relatively short. Moreover, the surface topography of mountain glaciers is often highly uneven and the weather stations on them are usually set up at their lower parts. Thus, the representativeness of these SAT
observations on glaciers is limited and the thermal status in their other parts is still unclear (Kang et al., 2022; Yang et al., 2014). As to the problem of representativeness, if there are several automatic weather stations in some glacierized basin, the temperature lapse rate (TLR) with respect to increasing elevation is introduced to interpolate station SAT observations to areas on glaciers without stations (Zhang et al., 2021; Rounce et al., 2020a). However, the TLR is unstable and variant in space and time (Li et al., 2013). So, the interpolated SATs via the TLR on the glaciers are full of nontrivial uncertainties
(Rounce et al., 2020b). As well known, gridded reanalysis (or simulation) data usually have a long span of time, and thus are often taken to address the problems of the short-term duration (or the nonexistence) of station SAT observations on glaciers (Munoz-Sabater et al., 2021; Harris et al., 2014; Hersbach et al., 2020). However, since the spatial extent of mountain glaciers are normally smaller than that of the grid (about several tens of kilometers), the spatial downscaling has to be performed on these gridded data to obtain the distribution of SAT on glaciers, such as through the TLR. In fact, it is reported
that in many studies that the quality of reanalysis data on the TP is dubious and thus the reliability of the downscaled SATs based on them should be worse (Li et al., 2013; Wang et al., 2019; You et al., 2010).

Given a strong association between SAT and LST, attempts have been made to convert satellite LSTs to satellite SATs (Zhang et al., 2016; Shen et al., 2020; Benali et al., 2012; Rao et al., 2019). The methods can be generally divided into four types. The first one is to construct a statistical relationship between SATs and LSTs as well as other ancillary variables

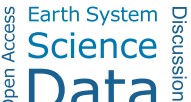

(such as solar zenith angle, elevation, etc.) via a multiple linear regression. This kind of methods are simple and can be calibrated with a few of data. But the relationship between SATs and LSTs are nonlinear and thus their reliability cannot be guaranteed in some situations. The second kind is called the temperature-vegetation-index method, which is based on the assumption that LST gradually approaches SAT with vegetation coverage increasing. This kind of methods are more accurate than the simple linear regression, but are susceptible to noise and inapplicable during the vegetation growing season.

The third kind of methods construct a quantitative relationship between SATs and LSTs through the surface energy balance equation, whose advantages are the solid physical basis. However, their accuracy depends strongly on the quality of the inputs, such as soil porosity, which are often difficult to be obtained. The fourth one exploits the strong ability of machine learning to capture the nonlinear relationship between SATs and LSTs as well as other ancillary variables (Xu et al., 2018; Noi et al., 2017; Zeng et al., 2021; Hooker et al., 2018). This kind of methods are successfully applied in many regions due

to its high accuracy and usability. The machine learning method has also been used to estimate SATs based on satellite LSTs over the TP (Shen et al., 2020; Xu et al., 2018; Zhang et al., 2016). However, the accuracy of these SATs on glaciers is questionable because the data-driven method needs a huge amount of data samples to train them but the amount of station SAT observations on glaciers are rather limited. Furthermore, satellite LSTs of high quality are only available for recent two decades. Even if the high-quality SATs were estimated on the TP, it would be difficult to use them to analyze climate change

on glaciers due to their short-term duration.

In order to address the aforementioned problems, we firstly glean SAT observations from tens of stations on glaciers of the TP, secondly construct a quantitative relationship between station SATs and satellite SATs to obtain short-term glacial SAT estimates through an ensemble learning algorithm, thirdly develop a reconstruction algorithm to temporally extend these satellite SATs to long-term (1961-2020) glacial SATs via the Bayesian linear regression, and finally implement

these reconstructed SATs to analyze the glacial warming trends on the TP for illustrating their utility. This article is organized as follows. In Section 2, the data sources are shown. Both the method to estimate satellite SATs and the approach to extend them are described in Section 3. The results and discussion are given in Section 4. Finally, the data availability and the conclusions are presented.

## 2 Data

### 2.1 Remote sensing data

A variety of Terra and Aqua MODIS products are used in this study, which are available from the National Aeronautics and Space Administration (NASA) website (https://earthdata.nasa.gov/). The first is the MOD11A1/MYD11A1 product, which provides the daytime and nighttime 1-km LST, satellite overpass time, and quality control indicators for each pixel. Its temporal resolution is one day and the spatial resolution is about 1 km. The second is the Enhanced Vegetation Index (EVI)

extracted from the MOD13A3 product. Its temporal and spatial resolutions are one month and 1 km, respectively. The third

is the shortwave white-sky albedo from the MCD43A3 product, whose temporal and spatial resolutions are one day and 500 m, respectively.

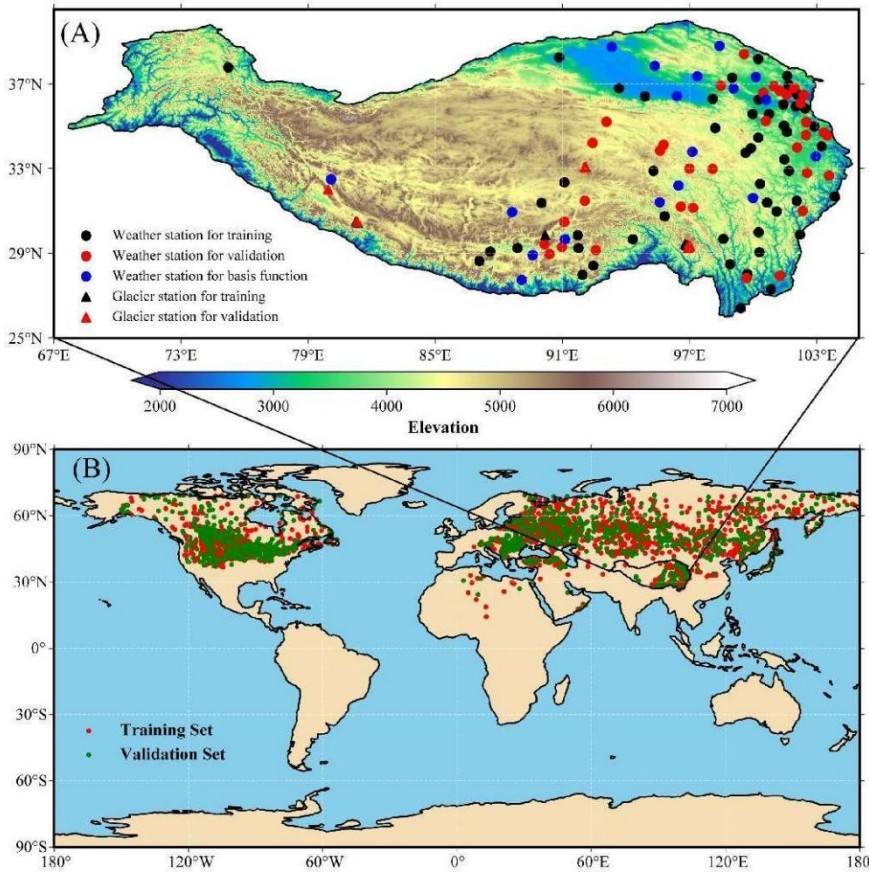

**Figure 1. Distribution of used ground observation stations.** (a) Meteorological stations from the CMA (black) and stations on glaciers (red); (b) Meteorological stations from the National Centers for Environmental Information of the United States.

### 2.2 Station products and ancillary data

The station SAT observation datasets used in this study come from three sources. The first one is the 60 years of daily near-surface air temperatures at a total of 145 weather stations on the TP, which are managed by the China Meteorological

Administration (CMA) and available from its website (http://data.cma.cn/), whose spatial distribution is shown in Fig. 1a (marked by the solid circles). These daily SATs are averaged to obtain monthly ones that match the MODIS monthly LSTs on the time scale. The second one is the glacial SAT observations at 35 automatic weather stations deployed on various glaciers of the TP (marked by the solid triangles), including the Kunsha Glacier in the western part of the TP(Yinsheng, 2018b), the Naimona'Nyi Glacier in the southwestern part of the TP(Yinsheng, 2018a), the Xiao Dongkemadi Glacier in the

central part of the TP(Baiqing, 2018), and several glaciers in the southeastern part of the TP(Wei, 2021). As aforementioned,

a machine learning model is taken to convert LSTs to SATs. If the amount of data is insufficient, the problem of overfitting is likely to occur. In order to mitigate this possible issue, a large number of weather stations in the Northern Hemisphere (outside of the TP), which have similar surface conditions to the TP, are selected to enrich the above two data sources. The selection criterion is that the annual mean albedo at these stations is greater than 0.4 because the land on the TP is typically

covered by sparse vegetation and ice (snow). The spatial distribution of these selected stations outside of the TP are illustrated in Fig. 1b and available from the National Center for Environmental Information of the United States (https://www.ncei.noaa.gov/). The most conspicuous feature of the TP is its extremely high elevation. So, the digital elevation model (DEM) data with a spatial resolution of 30 meters, which are from the Space Shuttle Radar Topography Mission (SRTM), is used to reflect this feature. This DEM data can be available from the U.S. Geological Survey website

(https://srtm.csi.cgiar.org/). The extent of glaciers on the TP comes from the Randolph Glacier Inventory provided by the National Tibetan Plateau Data Center of China (http://data.tpdc.ac.cn/).

## 3 Methods

Suspendisse a elit ut leo pharetra cursus sed quis diam. Nullam dapibus, ante vitae congue egestas, sem ex semper orci, vel sodales sapien nibh sed lectus. Etiam vehicula lectus quis orci ultricies dapibus. In sit amet lorem egestas, pretium sem sed,

tempus lorem.

### 3.1 Data preprocessing

Since the purpose of this study is to estimate and reconstruct monthly SATs on glaciers of the TP, both the daily station SAT observations and satellite LSTs are averaged to obtain monthly ones. For weather and glacier stations, all daily SAT observations are simply averaged in one calendar month. As to MODIS LST products, the averaging procedure is a little

complicated. It is well known that satellite signals are often contaminated by clouds, leading to no LST retrievals or low-quality ones. So, the quality control is conducted on the four daily MODIS LSTs (Terra daytime LST, Terra nighttime LST, Aqua daytime LST, and Aqua nighttime LST) and only the LSTs, whose uncertainties are less than 1K, are regarded as valid. Then, the averaging procedure is only performed on these LST values for each 1-km pixel on the TP. Therefore, four monthly MODIS LSTs are calculated. At the same time, the total number of valid daily LSTs in one month is used to reflect

the cloud information for one pixel and normalized into the range of 0–1. Moreover, four monthly mean satellite overpass times are computed in the same manner and then normalized into the range of 0–1 through being divided by the length of one day (24 hours). As to MODIS 500-m daily albedos, they are aggregated into monthly scale by simply averaging and then resampled to 1-km spatial scale in order to match with the spatial and temporal scales of the other satellite products. For the same reason, the USGS 30-m DEM data are also resampled to 1-km spatial scale on the TP. As a matter of fact, the DEM

data is not used directly but converted into normalized air pressure belonging to the range of 0–1 as follows:



$$\eta = 10^{\left(\frac{-Z}{18400} \cdot \frac{LST}{273}\right)}, \tag{1}$$

where $\eta$ denotes the normalized air pressure, $Z$ the altitude above sea level (i.e., DEM), and $LST$ the land surface temperature. As pointed out in many studies, both sunrise and sunset times have an impact on the maximum and minimum SATs and their timing. So, these two times are calculated on a daily basis, the monthly averaged ones are simply evaluated

in a calendar month, and added into the list of input variables for the ensemble learning method.

### 3.2 Method to convert LSTs to SATs

As aforementioned, a stacking ensemble learning algorithm is constructed to estimate short-term satellite SATs and then Bayesian linear regression is used to reconstruct long-term glacial SATs. The whole procedure is illustrated in Fig. 2. For estimation, random forest model is taken as the base learner and meta learner for ensemble learning. Random forest

regression is a supervised machine learning algorithm to merge multiple regression trees to make a more accurate prediction than any individual tree. Its key thought is to perform bootstrap aggregation in both sample and feature dimensions in the learning course. The base learner can be represented as:

$$SAT = RF(X), \tag{2}$$

where $SAT$ denotes the monthly mean SATs and $X$ the independent variable vector. There are four base learners,

corresponding to four MODIS LST, respectively, and thus there are four input vectors, which can be expressed as follows:

$$
\begin{aligned}
X_{day}^{terra} &= \left[ LST_{day}^{terra}, \tau_{day}^{terra}, t_{day}^{terra}, p_{day}^{terra}, vi, \alpha, t_{sunrise}, t_{sunset}, \cos\theta_{sun} \right] \\[6pt]
X_{night}^{terra} &= \left[ LST_{night}^{terra}, \tau_{night}^{terra}, t_{night}^{terra}, p_{night}^{terra}, vi, \alpha, t_{sunrise}, t_{sunset}, \cos\theta_{sun} \right] \\[6pt]
X_{day}^{aqua} &= \left[ LST_{day}^{aqua}, \tau_{day}^{aqua}, t_{day}^{aqua}, p_{day}^{aqua}, vi, \alpha, t_{sunrise}, t_{sunset}, \cos\theta_{sun} \right] \\[6pt]
X_{night}^{aqua} &= \left[ LST_{night}^{aqua}, \tau_{night}^{aqua}, t_{night}^{aqua}, p_{night}^{aqua}, vi, \alpha, t_{sunrise}, t_{sunset}, \cos\theta_{sun} \right]
\end{aligned} \tag{3}
$$

Here, the first four terms in the four input vectors ( $LST$, $\tau$, $t$, and $p$ ) denote the MODIS land surface temperature, the clear-sky days, the satellite overpass time, and the air pressure during the daytime and nighttime for the Terra and Aqua satellites. The other five terms in the vectors ( $vi$, $\alpha$, $t_{sunrise}$, $t_{sunset}$, and $\cos\theta_{sun}$ ) represent the MODIS enhanced

vegetation index, the MODIS surface albedo, the sunrise time, the sunset time, and the cosine of sun zenith angle, respectively. Four estimated monthly surface air temperatures $\mathbf{SAT}^{est} = \left[ SAT_{day}^{terra}, SAT_{night}^{terra}, SAT_{day}^{aqua}, SAT_{night}^{aqua} \right]^{T}$ can



be procured by plugging the four input vectors ( $X_{day}^{terra}$, $X_{night}^{terra}$, $X_{day}^{aqua}$, and $X_{night}^{aqua}$ ) into Eq. (2). The combined surface air temperature $SAT^{com}$ can be computed through the meta learner as:

$$SAT^{com} = RF\left(\mathbf{SAT}^{est}\right). \tag{4}$$

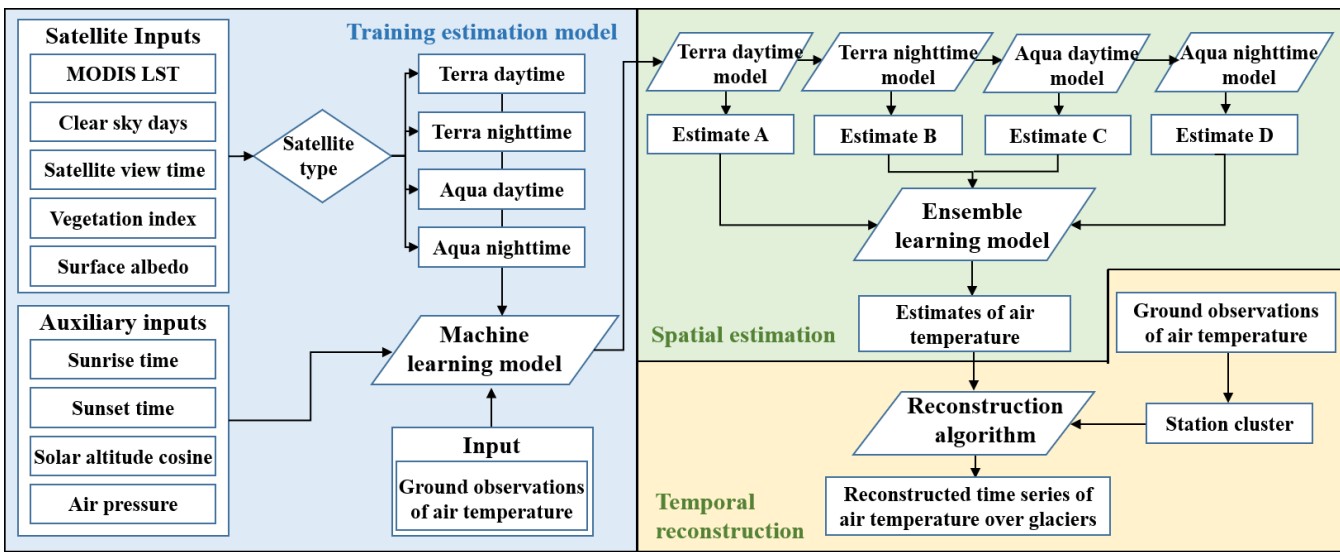


**Figure 2. Procedure to reconstruct 60-year (1961–2020) near-surface air temperature on glaciers**

### 3.3 Temporal reconstruction algorithm

After the satellite SATs have been retrieved by the above ensemble learning algorithm, the temporal extension is performed for each glacial pixel on the TP based on Bayesian linear regression and can be expressed as:

$$SAT_{h,i}^{ext} = \boldsymbol{\beta}_i^{\mathrm{T}} \bullet \mathbf{SAT}_h^{obs}, \tag{5}$$

where $h$ denotes the time index for the historical period of 1961–2020, $SAT_{h,i}^{ext}$ the reconstructed historical SATs at pixel $i$,

$\mathbf{SAT}_t^{obs} = \left[1, SAT_{t,1}^{obs}, SAT_{t,2}^{obs}, \ldots, SAT_{t,N}^{obs}\right]^{\mathrm{T}}$ the monthly mean surface air temperature observation at $N$ basis stations,

and $\boldsymbol{\beta}_i$ the vector of extension coefficients. The extension weights are estimated by minimization of the following cost function:

$$J = \sum_{t=1}^{T_i} \left(SAT_{t,i}^{com} - \boldsymbol{\beta}_i^{\mathrm{T}} \cdot \mathbf{SAT}_t^{obs}\right)\sigma_i^{-2}\left(SAT_{t,i}^{com} - \boldsymbol{\beta}_i^{\mathrm{T}} \cdot \mathbf{SAT}_t^{obs}\right) + \lambda_i \boldsymbol{\beta}_i^{\mathrm{T}} \boldsymbol{\beta}_i, \tag{6}$$




where $t$ denotes the month in the period of 1961–2020, $SAT_{t,i}^{com}$ the combined SAT for pixel $i$ at time $t$, $\sigma_i$ the standard deviation of $SAT_{t,i}^{com}$, $\lambda_i$ the regularization parameter to avoid overfitting, and $T_i$ the number of the estimated SATs for pixel $i$. The specification of $\sigma_i$ and $\lambda_i$ has a definitive impact on estimate of $\boldsymbol{\beta}_i$. Here, the variational Bayes method is utilized to optimize the cost function $J$ in order to simultaneously estimate $\boldsymbol{\beta}_i$, $\sigma_i$, and $\lambda_i$. The readers are referred to

the article for more details (Qin et al., 2013).

### 3.4 Evaluation indicators

In this study, three error metrics are selected to validate the glacial SAT estimates, which are root mean square error (RMSE), mean bias error (MBE) and determination coefficient ($R^2$). They are expressed as follows:

$$
\begin{aligned}
\text{RMSE} &= \sqrt{\frac{1}{M}\sum_{i=1}^{M}\left(y_i^{est}-y_i^{obs}\right)^2} \\
\text{MBE} &= \frac{1}{M}\sum_{i=1}^{M}\left(y_i^{est}-y_i^{obs}\right) \\
R^2 &= 1-\frac{\sum_{i=1}^{M}\left(y_i^{est}-y_i^{obs}\right)^2}{\sum_{i=1}^{M}\left(y_i^{est}-\overline{y}\right)^2} \\
\overline{y} &= \frac{1}{M}\sum_{i=1}^{M}y_i^{obs}
\end{aligned} \qquad (7)
$$

where $y^{est}$ denotes the estimated SATs (including estimated satellite SATs and reconstructed glacial SATs), $\overline{y}$ the averaged station SAT observations, and $M$ the number of stations observations. When the evaluation is performed according to RMSE, MBE, and $R^2$, it often happens that RMSE, MBE, and $R^2$ exhibit inconsistency. For example, RMSE of one dataset is larger than RMSE of the other dataset, but $R^2$ is also larger than $R^2$ of the other dataset. Here, a comprehensive error metric (DISO) is introduced to handle this and can be formulated as Zhou et al., 2021):

$$\text{DISO}=\sqrt{\left(R-1\right)^2+\left(RMSE/\overline{y}\right)^2+\left(RMSE/\overline{y}\right)^2}\,, \qquad (8)$$

# 4 Results and discussion

## 4.1 Evaluation of satellite SATs

Figure 3A shows the overall training results for the presented ensemble learning algorithm. There are a total of 17901 samples from July 2002 to December 2020 in the training dataset (in synchronization with the time span of MODIS). As can be seen, the training accuracy can attain an extremely high level with RMSE, MBE, and $R^2$ being equal to 0.17 °C, 0.00 °C, and 1.00, respectively. This indicates the strong capability of the ensemble leaning algorithm to capture the variation in the samples. As a matter of fact, the validation results should be more concerned. Figure 3B shows the overall validation results with RMSE, MBE, and $R^2$ being equal to 1.47 °C, 0.11 °C, and 0.98, respectively. Although the overall validation accuracy is inferior to the overall training accuracy, the validation results are fairly favorable. Moreover, the validation results are illustrated on three groups of stations in Fig. 4. As shown in Fig. 4A, the validation results at global regular stations outside of the TP are similar to the overall validation results with RMSE, MBE, and $R^2$ being equal to 1.60 °C, -0.03 °C, and 0.97. For regular stations in the TP, RMSE, MBE, and $R^2$ are equal to 1.35 °C, 0.11 °C, and 0.97, respectively. For only glaciers, the validation results are slightly worse than those at regular stations, with RMSE, MBE, and $R^2$ being equal to 1.61 °C, 0.21 °C, and 0.93. The total number of glacier stations for validation is 18 and the validation results for each station are listed in Table 1. As can be seen, most of glacier stations are set up in 2018 and their observation periods last for less than two years. Moreover, the number of high-quality monthly SAT observations are rather limited and only one observation sample is available at two stations (SETP12 and SETP13) due to harsh natural conditions. Only SETP1 has a long observation period lasting for approximately 15 years.

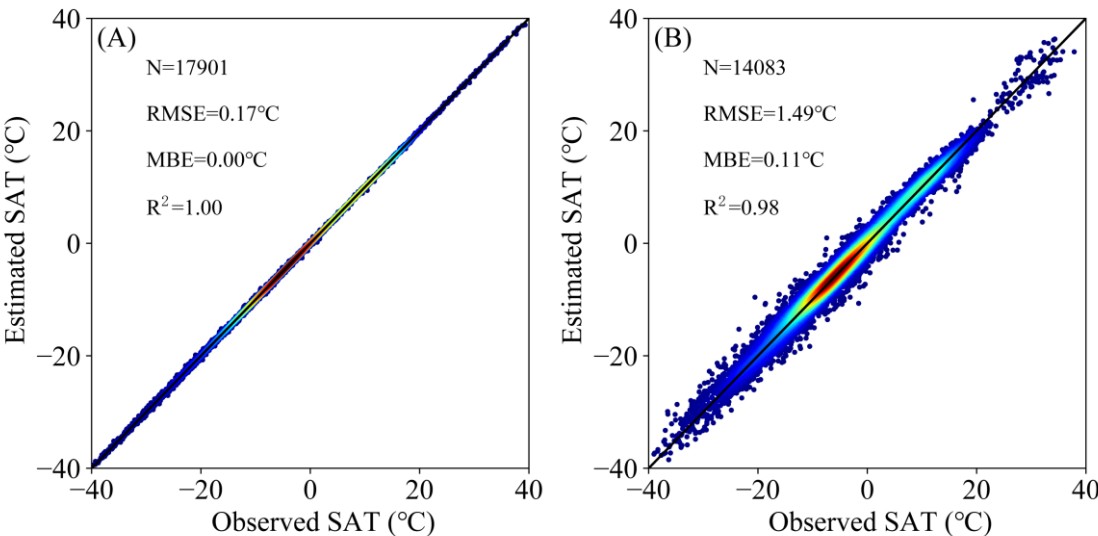

**Figure 3. Comparison of observed station SATs with estimated satellite SATs (A) for training and (B) for validation on the whole training and validation datasets.**



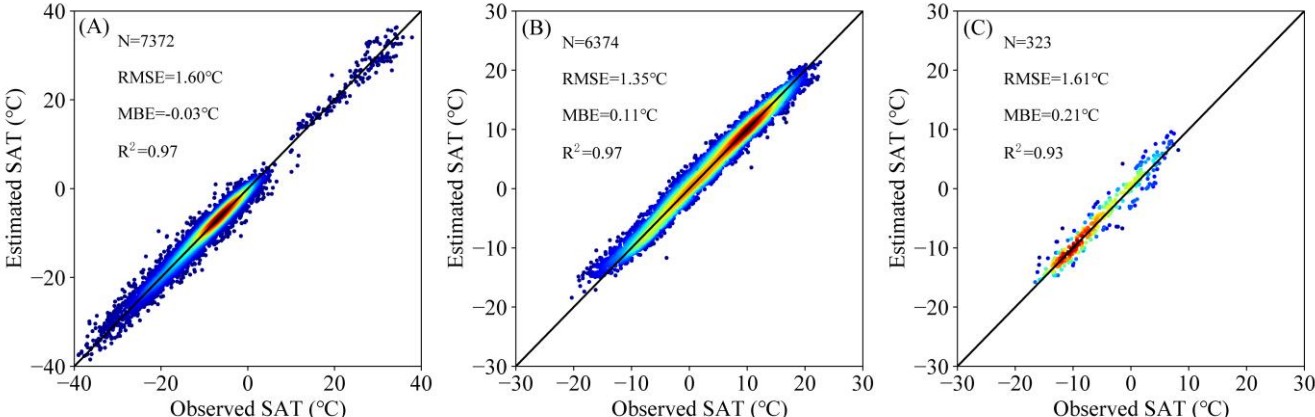

**Figure 4. Comparison of observed station SATs with estimated satellite SATs for validation (A) at regular stations in the northern hemisphere beyond the Tibetan Plateau, (B) at regular stations on the Tibetan Plateau, and (C) at glacier stations.**

**Table 1 validation results for each glacier station on the TP**

| Station | Latitude (°) | Longitude (°) | Elevation (m) | RMSE (°C) | $R^2$ | MBE (°C) | Number of samples | Start and end time |
|---|---|---|---|---|---|---|---|---|
| SETP1 | 29.3184 | 96.9563 | 4602 | 1.495 | 0.94 | 0.638 | 125 | 2006.07-2020.10 |
| SETP2 | 29.3481 | 97.0227 | 5095 | 2.956 | 0.793 | -0.73 | 7 | 2018.07-2019.09 |
| SETP3 | 29.352 | 97.0209 | 5168 | 2.788 | 0.821 | -0.09 | 7 | 2018.07-2019.09 |
| SETP4 | 29.355 | 97.0202 | 5258 | 2.833 | 0.826 | 0.115 | 7 | 2018.07-2019.09 |
| SETP5 | 29.3568 | 97.0201 | 5310 | 1.231 | 0.951 | 0.393 | 8 | 2018.07-2019.09 |
| SETP6 | 29.3576 | 97.0194 | 5335 | 1.349 | 0.943 | 0.779 | 8 | 2018.07-2019.09 |
| SETP7 | 29.4133 | 96.9661 | 4965 | 0.423 | 0.996 | -0.294 | 7 | 2018.07-2019.09 |
| SETP8 | 29.3961 | 96.9726 | 5138 | 0.554 | 0.991 | 0.135 | 4 | 2018.07-2019.09 |
| SETP9 | 29.3939 | 96.9727 | 5174 | 1.486 | 0.95 | -0.57 | 6 | 2018.07-2019.09 |
| SETP10 | 29.3883 | 96.9701 | 5302 | 1.472 | 0.951 | -0.065 | 6 | 2018.07-2019.09 |
| SETP11 | 29.3864 | 96.9735 | 5280 | 1.586 | 0.943 | -0.544 | 6 | 2018.07-2019.09 |
| SETP12 | 29.3142 | 96.9557 | 4588 | 0.564 | / | -0.564 | 1 | 2018.07-2019.09 |
| SETP13 | 29.265 | 96.9379 | 4649 | 1.118 | / | 1.118 | 1 | 2018.07-2019.09 |
| SETP14 | 29.2436 | 96.9258 | 4909 | 0.564 | 0.99 | -0.074 | 10 | 2018.07-2019.09 |
| CTP1 | 32.0191 | 79.9505 | 5100 | 1.4499 | 1.267 | 0.9506 | 17 | 2015.10-2017.09 |
| SWTP1 | 33.0571 | 92.0525 | 5255 | 1.194 | 0.96 | 0.272 | 7 | 2018.11-2019.10 |
| SWTP2 | 33.0689 | 92.0733 | 5627 | 1.207 | 0.953 | 0.581 | 19 | 2012.05-2015.08 |
| CTP2 | 30.4922 | 81.3138 | 5543 | 1.794 | 0.866 | -0.89 | 77 | 2011.10-2018.11 |

To further examine the reliability of the satellite SAT estimates on glaciers, comparisons with three other SAT datasets are performed. Two of them are the satellite SAT datasets based on MODIS LST products with machine learning methods (Xu et al. 2018; Chen et al. 2021), which cover the periods of 2001–2015 and 2001–2019, respectively, with a

spatial resolution of 1km×1km. As can be seen in Fig. 5, Xu's SAT product has RMSE, MBE, and $R^2$ of 3.11 °C, 0.84 °C, and 0.81, respectively, over the glaciers. However, three error metrics for our glacial satellite SAT product are equal to

1.34 °C, -0.13 °C, and 0.96, obviously being superior to those for Xu's SAT products. As to Chen's product, the error metrics are 3.67 °C, 1.19 °C, and 0.67, respectively, being inferior to ours as indicated in Fig. 6.

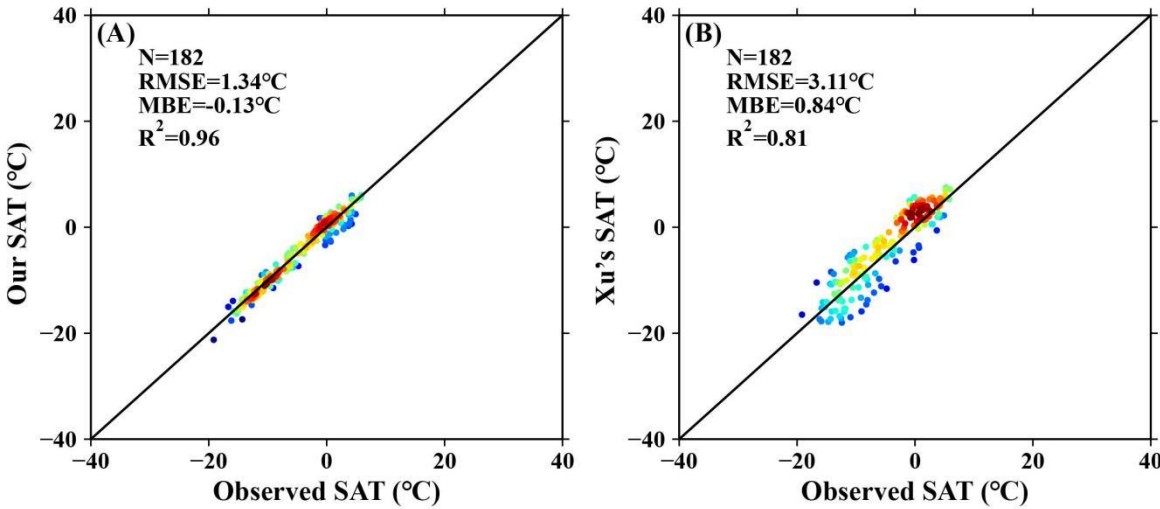

**Figure 5. Comparison between station SAT observations and satellite SAT estimates over glaciers for (A) our products and (B) Xu's products Xu et al., 2018).** There is a total of 182 sample points since the overlay period of these two datasets are from 2002 to 2015.

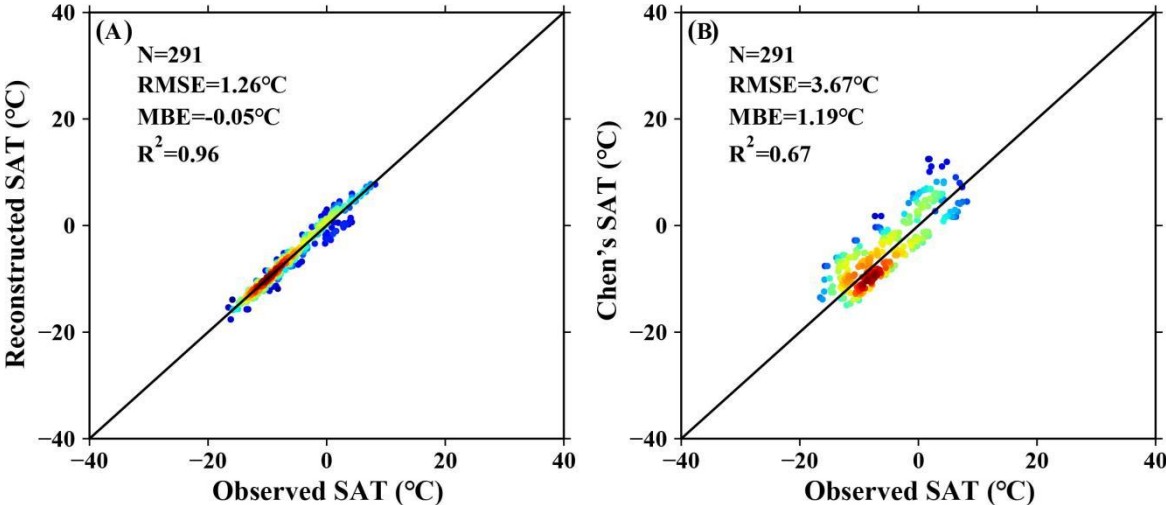

**Figure 6. Comparison between station SAT observations and satellite SAT estimates over glaciers for (A) our products and (B) Chen's products (Chen et al., 2021).** There is a total of 291 sample points since the overlay period of these two datasets are from 2002 to 2019.

The other is the ERA5-Land reanalysis dataset. It has a spatial resolution of 0.1° and a time span from 1951 to the present. As can be seen in Fig. 7A and C, the performance of our SAT products significantly surpasses the one of ERA5-Land since RMSE, MBE, and $R^2$ deteriorate from 1.32 °C, -0.31 °C, and 0.96 to 3.60 °C, -0.71 °C, and 0.81. At the same



time, the satellite SAT estimates at a 1 km×1 km resolution are aggregated to the ones at a 0.1° resolution and then compared

with the corresponding ERA5-Land products in order to investigate whether or not the inferiority of ERA5-Land SATs over the glaciers is caused by the difference in the spatial scale. It is illustrated in Fig. 7B that the aggregated SATs are also superior to the ERA5-Land ones over the glaciers with RMSE, MBE, and $R^2$ equal to 2.46 °C, -0.52 °C, and 0.88. This proves the advantage of the presented satellite SAT retrieval algorithm itself after adding the samples over glaciers into estimation model.

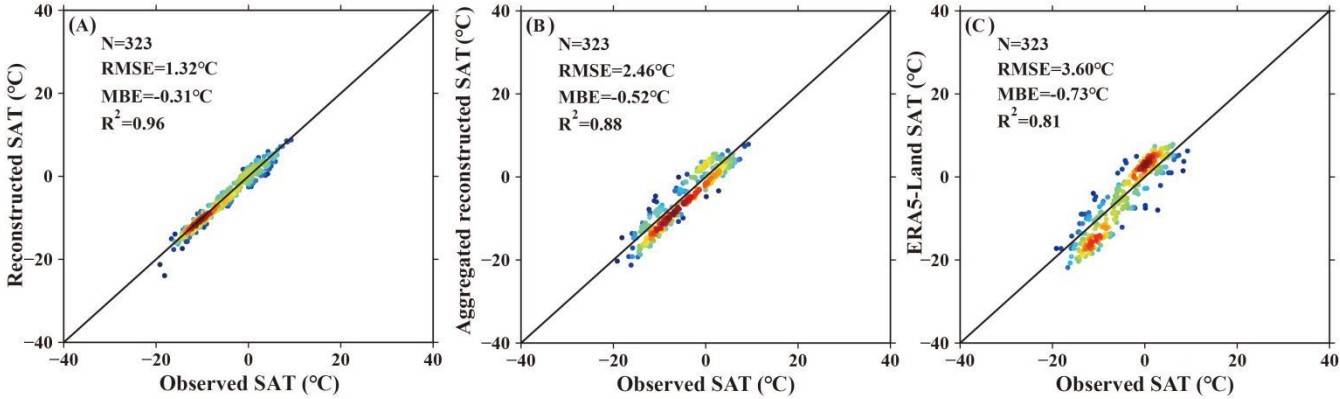


**Figure 7. Comparison between station SAT observations and satellite SAT estimates over glaciers for (A) our SAT products at a spatial resolution of 1 km×1 km, (B) our products aggregated to a resolution of ERA5-Land 0.1°, and ERA5-Land SATs.** There is a total of 323 sample points since the overlay period of these two datasets are from 2002 to 2020.

**4.2 Evaluation of reconstructed glacial SATs**

The primary goal of this study is to reconstruct long-term glacial SATs to deepen our understanding of the glacial warming status on the TP. As mentioned in subsection 3.3, long-term continuous SAT observations at some stations need to be selected as basis functions. In principle, the more the number of basis functions, the better the reconstructed results. In fact, the number of such basis functions are scant because harsh natural conditions often cause failure of measuring instruments and thus observations miss. So, a total of 18 regular weather stations, at which there are no missing observations in 1961-

2020, are selected out of 145 stations as basis functions (Fig. 8). It is notable that the temporal extension is performed on the monthly basis and the annually averaged values are just presented for illustration. As can be seen, the spatial variabilities of these basis functions are rather striking (Fig.1), but the temporal variabilities show similarities to some degree. For example, the SAT differences between stations 52062 and 56034 can attain more than 12.0 °C due to their distinct elevations and long distance. Before the reconstruction is performed for every glacier pixel, five ideal experiments are conducted at the pixel,

where the glacier station SETP1 is located, in order to substantiate the efficacy of the reconstruction algorithm. The procedure for each experiment consists of three steps. Firstly, observation period (2006–2020) is partitioned into two parts. Secondly, the glacial satellite SAT estimates in the later part are used to evaluate the extension coefficients $\boldsymbol{\beta}$ in Eq. (5). Thirdly, the glacial station SAT observations in the earlier part are used to validate the reconstructed SATs. The difference among these five experiments lies in the various years to separate the observation period. The separation years are 2013,





2011, 2009, 2007, and 2005, respectively. As can be seen in Fig. 9A1, B1, C1, D1, and E1, the number of satellite SAT estimates, which are used to compute the extension coefficients, increases with the separation year moving forward and accordingly the number of station SAT observations declines. At the same time, the validation accuracy (RMSE, MBE, and $R^2$) gradually improves in the first four experiments as shown in Fig. 9A2, B2, C2, and D2. It is notable that no validation is performed for the fifth experiment since the separation year of 2005 means that all satellite SAT estimates in the observation

period are used in evaluating the extension coefficients.

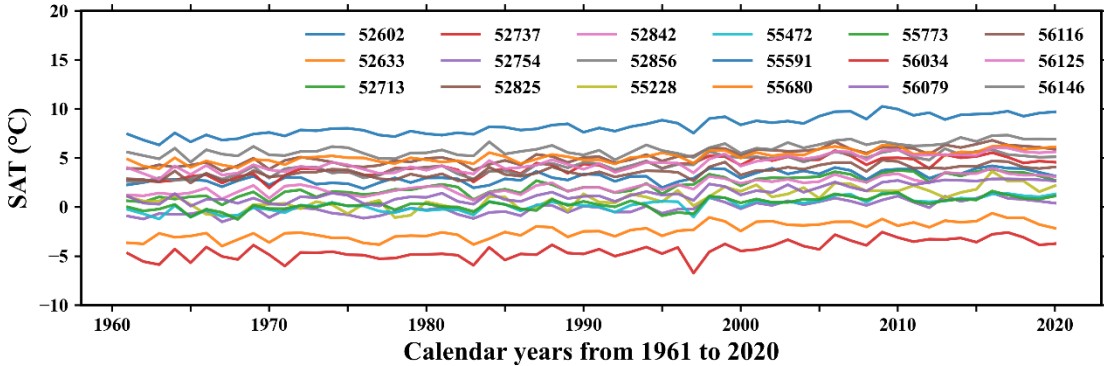

**Figure 8. Surface air temperature time series from 1961 to 2020 at 18 stations selected as basis functions for reconstruction.**



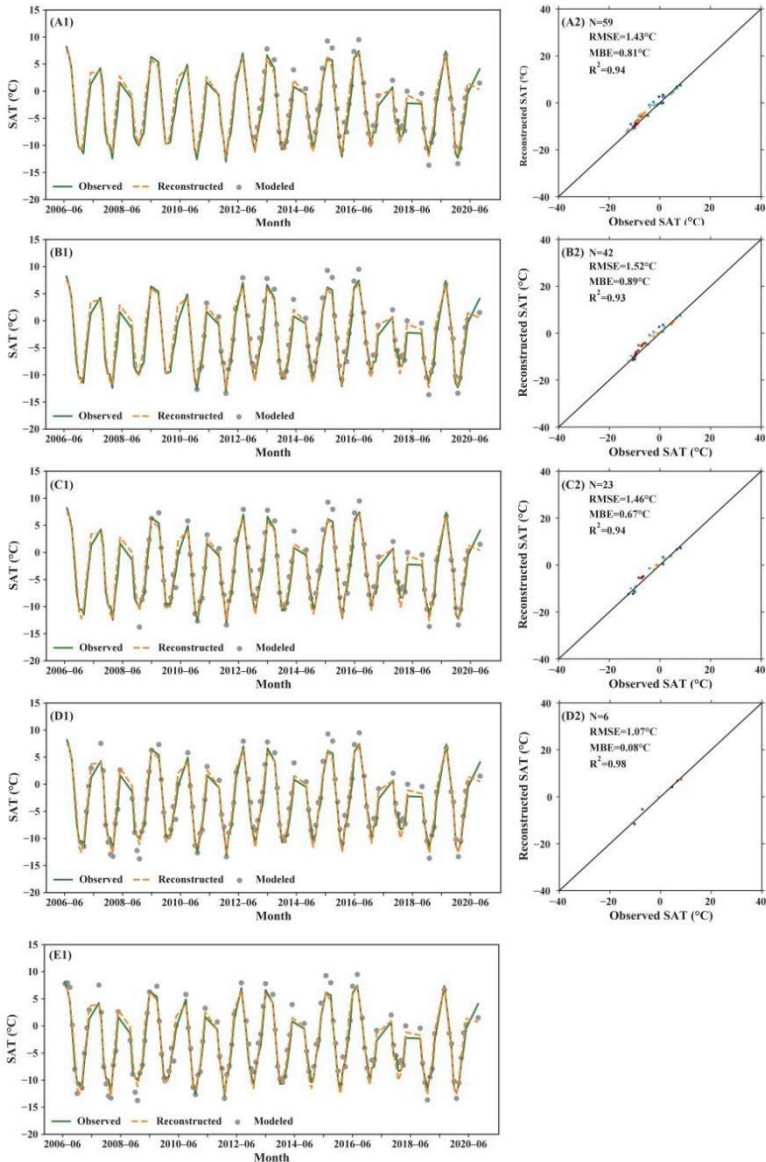

**Figure 9. Validation of reconstructed SATs at glacier station SETP1 in five ideal experiments with various specified periods in which satellite SAT estimates are available.** (A) for the period of 2013 to 2020, (B) for the period of 2011 to 2020, (C) for the period of 2009 to 2020, (D) for the period of 2007 to 2020 and (E) for the period of 2005 to 2020.

As a matter of fact, in the five ideal experiments, the temporal extension process can produce not only the reconstructed SATs in the time period in which no satellite SAT estimates exist, but also the ones in the period when satellite SATs are available. These reconstructed SATs in the latter period can be regarded as being smoothed. Figure 10A–E illustrate the comparison between all reconstructed SATs and observed SATs for the five ideal experiments. As shown in Fig. 10F, the comprehensive error metric (DISO) becomes better with the number of satellite SAT estimates rising. The ideal



experiments are performed only at one glacier station with a long-term series of observations and such experiments cannot be done at other glaciers stations due to the lack of enough SAT observations. However, these ideal experiments demonstrate that the presented reconstruction algorithm can restore the historical SATs on glaciers with high accuracy.

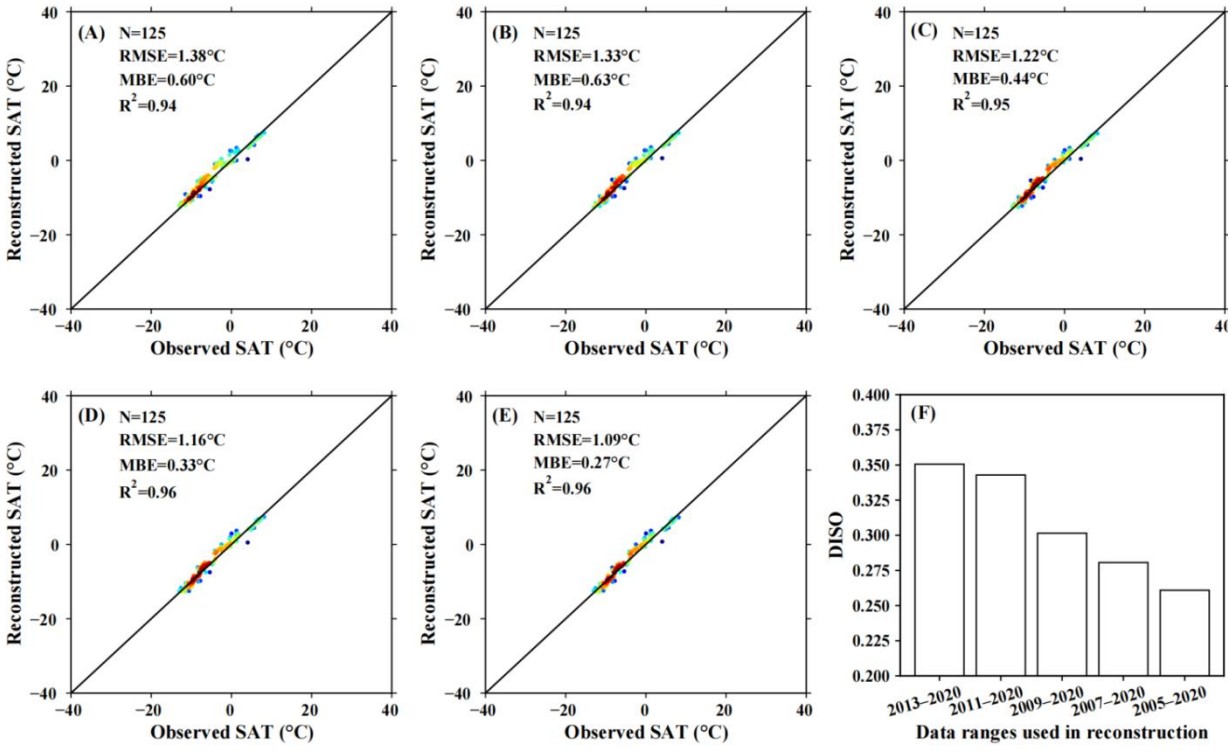


**Figure 10. Comparison between observed SATs and reconstructed SATs in the period of 2006 to 2020 at glacier station SETP1 in five ideal experiments with various specified periods in which satellite SAT estimates are available.** (A) for the period of 2013 to 2020, (B) for the period of 2011 to 2020, (C) for the period of 2009 to 2020, (D) for the period of 2007 to 2020, (E) for the period of 2005 to 2020, and (F) the error metric DISO in the five experiments.

## 4.3 Glacial warming pattern

Both annually means and anomalies of reconstructed SATs at each glaciated pixel are calculated for the period of 1961 to 2020. Then, the anomalies are averaged over all glaciated pixels to obtain the holistic warming trend of the glaciated area on the TP. As can be seen in Fig. 11A, the glaciated areas on the TP have undergone a rapid warming in the past 60 years with a warming rate of 0.024 °C/yr at the 95% confidence level. It is also found that the coldest and warmest SATs occur in 1967 and 2016 with averaged values of -11.20 °C and -9.33 °C, respectively. In order to obtain a complete understanding of the warming pattern in space, the warming trends for each glaciated pixel are illustrated in Fig. 11B. Except for a few glaciers in the southeast, the warming happens over almost all glaciers on the TP, but at the same time exhibits a spatial heterogeneity, being more pronounced in the north. The maximum warming rate reaches 0.07°C/yr, appearing over the glaciers in the central Karakorum Mountains located in the northwestern part of the TP. The cooling occurs over the glaciers of the Western



Himalayas in the southwest of the TP and the glaciers of the Nyainqentanglha Mountains in the south of the TP, but the cooling trends are not significant.

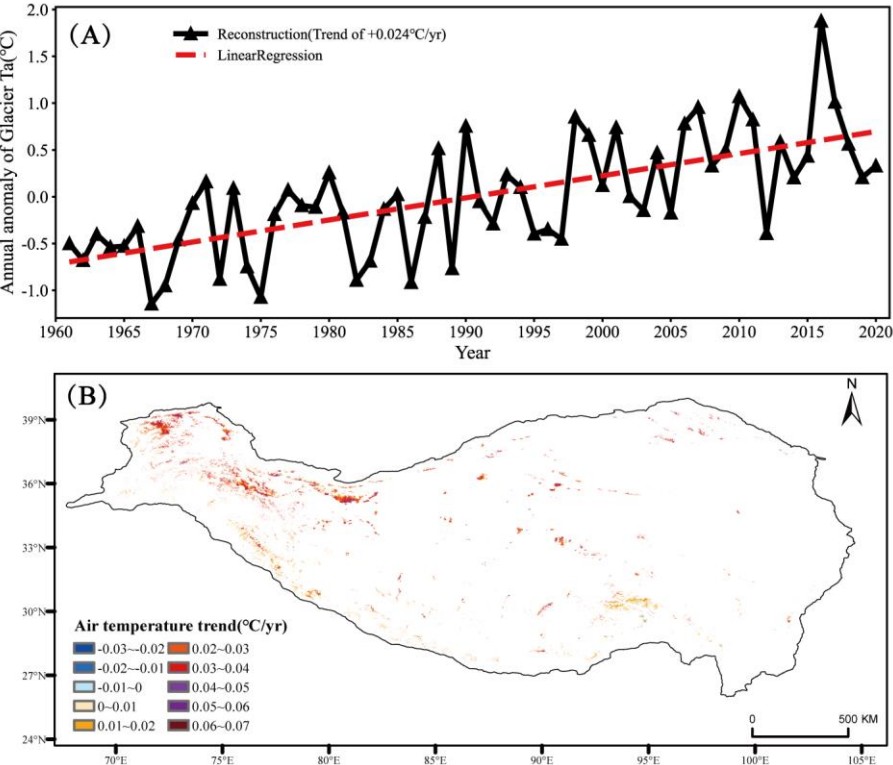

**Figure 11. Warming trends over glaciers of the Tibetan Plateau from 1961–2020.** (A) Annual anomalies of surface air temperatures over all glaciers and. (B) Spatial pattern of glacial warming trends.

## 5 Data availability

The 60-years (1961–2020) of glacial near-surface air temperature dataset on the Tibetan Plateau is freely available from the National Tibetan Plateau Data Centre at https://doi.org/10.11888/Atmos.tpdc.272550 (Qin 2022). The dataset provides monthly estimates of near-surface air temperature within 67.67°E–104.67°E and 26.01°N–40.00°N at a spatial resolution of 1 km in units of °C. All files are stored in GeoTIFF format with a datum of WGS84. Each file is named as "yyyymm.tif", where "yyyy" and "mm" denote year and month, respectively. For example, the file "196101.tif" stores the glacial monthly near-surface air temperature on the Tibetan Plateau in January 1961.

## 6 Summary

The shortage of long-term glacial SATs with high spatial resolution has seriously hindered the deep understanding of glacial warming status on the TP. On the basis of MODIS LST products and station SAT observations, we develop an ensemble

learning algorithm with a random forest being the base learner to convert MODIS LSTs to SATs over the glaciers of the TP
from 2002 to 2020. The glacial satellite SAT estimates are validated against glacial station SAT observations with RMSE,
MBE, and $R^2$ equal to 1.61 ℃, 0.21 ℃, and 0.93, respectively. At the same time, a series of experiments are conducted to
corroborate the effectiveness of the temporal extension algorithm. Afterwards, long-term SATs between 1961 and 2020 are
reconstructed for all the glacier pixels over the TP. Based on the reconstructed SATs, the warming trend from 1961 to 2020

over all the glaciers of the TP is equal to 0.024 ℃/yr in the past 60 years. The spatial warming pattern indicate that most of
the glaciers are undergoing a warming process with a maximum warming rate of 0.07 ℃/yr, and only a few glaciers in the
southeastern part of the TP exhibit a insignificant cooling trend. Overall, this study alleviates the problem of being short of
long-term SAT data over the glaciers of the TP. The reconstructed SAT dataset can strongly underpin climate change and
modelling researches on glaciers of the TP. In the future, we intend to enhance the temporal resolution of the glacial SAT

dataset to daily scale and implement the spatial extension outside of the TP to other ice-covered areas in the world, such as
the North and South Poles.

**Author contributions.** Jun Qin, Ning Lu, Ling Yao, and Chenghu Zhou designed the experiments. Jun Qin developed the
model code. Weihao Pan, Min He, and Hou Jiang performed the simulations. Jun Qin prepared the manuscript with
contributions from all co-authors.

**Competing interests.** The authors declare that they have no conflict of interest.

**Acknowledgements.** This study is jointly funded by the Third Xinjiang Scientific Expedition Program (Grant No.
2021xjkk0303) and Key Special Project for Introduced Talents Team of Southern Marine Science and Engineering
Guangdong Laboratory (Guangzhou) (GML2019ZD0301).  We are grateful to the National Tibetan Plateau Data Centre
(**http://data.cma.cn/**), the National Tibetan Plateau Data Center of China (**http://data.tpdc.ac.cn/**), and the National Center
for Environmental Information of the United States (**https://www.ncei.noaa.gov/**) for sharing the ground observation data.

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
