# Peer review of "A long-term 1-km monthly near-surface air temperature dataset over the Tibetan glaciers by fusion of station and satellite observations"

_Earth System Science Data, 2022_

## Author Comment (AC2)

**Response to Referee #2**

We would like to thank the reviewer for the comments and suggestions, which are all valuable and very helpful for improving our paper. We have made revisions and a point-to-point response is present in the following.

**Summary and comments:**

Qin et al. construct a 60–year (1961–2020) near-surface air temperature dataset over the glaciers of the Tibetan Plateau by fusing satellite and multi-source observations. The used ensemble learning model is described in detail and sufficient experiments are conducted to validate the reliability of constructed datasets. The manuscript is well organized, and all results are clearly presented.

**Response:**

We thank Referee #2 for the encouraging comments.

**Comments:**

Minor comments:

1) As the article is aimed at a data journal, I think it is better to include more key information on the dataset in the title of the manuscript, such as the spatial resolution (1km), temporal resolution (monthly) and used methods (data fusion or machine learning).

**Response:**

The authors have changed the title to "A long-term 1-km monthly near-surface air temperature dataset over the Tibetan glaciers by fusion of station and satellite observations".

**Comments:**

2) Section 3.2: It is necessary to present some reasons or your considerations for selecting the random forest model, by citing relevant literatures or adding concise discussions.

**Response:**

As the reviewer points out, there are many machine learning methods that could be selected to convert LSTs to SATs. There has already existed a study which compare a total of ten machine learning methods (including several methods mentioned by the reviewer) in converting LSTs to SATs. The result shows that Cubist and random forest rank the first two places and their difference is subtle. Moreover, the random forest has been successfully

applied in many studies to convert LSTs to SATs. Therefore, the authors have added description and citation "which has been proved effective in many scenarios (Belgiu and Dragut 2016; Xu et al. 2018)" in the revised manuscript.

**Comments:**

3) Eq. 8: Please give more description for DISO. How should readers interpret the value? Does a larger or smaller value mean better?

**Response:**

The authors have added "Overall, the smaller DISO is, the better estimates are." after Equation (8) to qualitatively describe the implication of DISO in the revised manuscript.

**Comments:**

4) Please add legends for all scatterplots.

**Response:**

The authors have modified all scatter plots in the revised manuscript by adding the color bars. These newly drawn figures are displayed in the following.

[Figure]

Figure 3.

[Figure]

Figure 4.

[Figure]

Figure 5.

Figure 6.

Figure 7.

**Comments:**

5) Whether the stations outside the TP in the Northern Hemisphere locate at glaciers? If yes, it is better to add a sub-figure in Figure 4 to show the validation results at such station.

**Response:**

These stations outside of the TP are not at glaciers. But we will try our best to collect records at glaciers, and we will consider more glacier stations all over the world the subsequent research.

---

## Author Response (AR1)

**Response to Reviewers**

Dear editor,

Thank you for giving us the opportunity to submit a revised draft of the manuscript "**A 60–year (1961–2020) near-surface air temperature dataset over the glaciers of the Tibetan Plateau**" for publication in Earth System Science Data. We appreciate the time and effort that you and the reviewers dedicated to providing feedback on our manuscript and believe that these comments are critical for improving this work. We have carefully considered and responded to the reviewers' comments and suggestions.

Note: the responses are colored as **blue**, the original comments from the reviewers are in black font, The modifications in the manuscript are colored as **red.**

**Reviewer #1:**

1. The monthly near-surface air temperature dataset with 1 km * 1 km spatial resolution during 1961-2020 over the glaciers of the TP is useful and important. The data and method are reasonable. The data especially satellite data used in this research are huge and the methods of big data processing (regression, machine learning algorithm) are close to the advanced international level. Evaluations and warming trend analysis results are acceptable. Logical analysis and English expression in the manuscript are good. The figures, tables and results are enough. This paper may cause higher impact factor with more citations, considering the high spatial resolution, long time-series and the special location.

2. The key problems in this research are well resolved and described in the manuscript. (1) Parameter: from LST to air temperature. (2) Temporal preprocessing: from day/night MODIS observation to monthly data, from 2002-2020 to 1961-2020. (3) Spatial preprocessing: from observations at hundreds of stations and satellite retrieval LST in clear sky with 1 km * 1 km to this dataset with 1 km * 1 km spatial resolution. (4) clear/cloudy circumstance. (5) Choose ERA-5 land to evaluate, and select the typical stations (shown in Figure 1A, in blue) to reconstruct air temperature time series as basis functions.

3.Major comments.

3.1 Discuss. Why not keep monthly and 1 km in the title? Over the glaciers of the TP, how about over the TP? The coverage of the dataset with 1 km resolution may not only over the glaciers. It is larger if it is over the TP, with all these meteorological weather stations including automatic stations and glacier stations. L32. How about add "with altitude more than 3500 m" after "highest plateau".

Response: (1) the reviewer's suggestion is very helpful. The authors have modified the title as "A long-term 1-km monthly near-surface air temperature dataset over the Tibetan glaciers by fusion of station and satellite observations". (2) The authors have completely understood the reviewer's concern. As a matter of fact, the reconstruction could be implemented on the entire TP for each 1-km grid. However, this work has not yet be conducted due to our limited computing resources. After the reconstruction work over the glaciers is published and there is a demand for the dataset over the TP, the authors will do it. (3) The suggestion for adding "with altitude more than 3500 m" is wonderful. The authors have corrected for this. The following words are added into the manuscript "with an average altitude greater than 3500 m" after "highest plateau" in L32.

3.2 L103 and L107. For the first one and the second one dataset used, do you choose homogenized products? If so, please add the description. If not, the authors should pre-process the data. The time series are long enough. Without homogenization processing, the trend results are not confident.

Response: the authors have totally understood the reviewer's concern. The authors have used the homogenized station dataset developed by Cao et al. The authors are sorry for neglecting this important information in the original manuscript and have added it into the revised manuscript as "The first one is the 60 years of daily near-surface air temperatures at a total of

145 weather stations on the TP, which are managed by the China Meteorological Administration (CMA), homogenized by Cao et al. (2016), and available from its website (http://data.cma.cn/), whose spatial distribution is shown in Fig. 1A (marked by the solid circles)......". At the same time, the corresponding article "Cao, L., Zhu, Y., Tang, G., Yuan, F., and Yan, Z.: Climatic warming in China according to a homogenized data set from 2419 stations, International Journal of Climatology, 36, 4384-4392, 2016." has been added into the revised manuscript.

3.3 L123-125, what are these?
Response: these characters have been caused in the process of converting the format. The authors are sorry for this. In the revised manuscript, they have been eliminated.

3.4 For clear-sky days, how to pre-process if it is than a certain criterion like 8 days or 5 days, or even 3 days? What is the criterion? Sometimes (what is the portion?) maybe the monthly LST results are missing in some 1 km pixels, the low-quality retrieval data are also not useful. The TP coverage of Terra and Aqua/MODIS is limited. The typical characters of LST in these months are kept missing or the interpolation from the adjacent two months?
Response: As the reviewer points out, monthly 1-km LSTs are sometimes missing. In order to mitigate this issue, the averaging is implemented once the number of high-quality LSTs at one pixel is greater than or equal to one. Of course, the authors know that the representativeness is rather weak when only a few LST values (even one value) in one month are used to calculate the mean. However, the authors think that data should not be discarded easily. Every data is informative and valuable. The suitable algorithm should be developed to handle the possible negative effect caused by this averaging procedure. So, Bayesian linear regression is taken to merge four estimated satellite SATs. Its most significant advantage is the ability to reduce the risk of being disturbed by points with large noises by adding a regularization term as shown in Equation (6). Moreover, the weight ($\lambda$) for this term can be estimated together with the regression coefficients. In reality, there have not existed any LST value in one month at a certain pixel and thus no monthly LST. As a matter of fact, it does not matter when this situation happens. As illustrated in the following figure, the basis functions (monthly mean SATs at 18 weather stations in this study) are continuous and their linear combination is fitted to the estimated satellite SATs. Thus, the reconstructed long-term (60 years) SATs are certainly continuous even though some estimated satellite SATs do not exit during 2002-2020. Bayesian linear regression is used to evaluate the combination coefficients ($\beta$). The authors believe that the above explanations could address the reviewer's concerns. In the revised manuscript, the sentence "Then, the averaging procedure is only performed on these LST values for each 1-km pixel on the TP" has been revised as "Then, the averaging procedure is only performed on these LST values for each 1-km pixel on the TP even though only one LST is available in one month." to give more details on the averaging procedure.

[Figure]

3.5 Why not choose GeoTIFF and ASCII format both?

Response: "GeoTIFF" format is a commonly used format and can be easily opened in many software. Moreover, data self-description can be realized. If the ASCII format is taken, the situation will become a little complicated. Besides the data files, a file have to be made, which describe the meta data (such as lon/lat for each grid) for the temperature dataset. However, the ASCII format is more visual for the data users. The authors will convert the format from GeoTIFF to ASCII and provide the dataset in these two types of formats.

4. Minor comments.

4.1 It is suggested to change the word "own".

Response: The authors have replaced the original word "owns" by the word "accommodates"

4.2 Figure 1 in L100. (a)(b), revise it to (A)(B). L100 and L105, 116. Figure 7 in Line 242, add (C). Figure 9 in L269, it is suggested to use A~I but not A2~D2 in the second column. Figure 9 is not clear, and the font size is too small.

Response: (1) The authors are sorry for these errors and have corrected them one by one. (2) It is a good advice to use A~I but not A2~D2 in Figure 9. Since, for example, the data in Figs. 9A1 and A2 are the same and just are displayed in two distinct ways, it can embody their internal logic connection to name them A1 and A2. Of course, if the reviewer think that it is more helpful for readers' understanding to use A~I, the authors will modify them according to the reviewer's comment in the next round revision. (3) The authors have revised Figure 9 according to reviewer's comments, enhancing the resolution and enlarging the font size.

4.3 Overfitting. Is it under fitting?

Response: Overfitting is a concept in data science, which occurs when a statistical model fits exactly against its training data. When this happens, the algorithm unfortunately cannot perform

accurately against unseen data. In this study, a machine learning algorithm (random forest) is taken to estimate near-surface air temperatures. Since its capacity is strong and the number of training samples is relatively scant, the overfitting easily happens. So, all weather stations in the northern hemisphere are used to increase the number of samples. In order to make this point clearer, the original words "**If the amount of data is insufficient, the problem of overfitting is likely to occur.**" have been modified as "Generally, this type of model has a strong capacity. Although approximately a few hundreds of stations are available over the TP, this number of stations is rather limited in contrast to the vast area of the TP. Thus, the problem of overfitting is likely to occur." in the revised manuscript.

4.4 L16 and L140. Is it near-surface air pressure? It is suggested to add "near-surface" in the abstract and the main body when it is the first time to mention.
Response: The authors are sorry for not clarifying this. The authors have added "near-surface" to L16 and L140 after "air pressure" in the revised manuscript.

4.5 Maybe "minimum and maximum" is better.
Response: the reviewer's comment is sensible. The words "maximum and minimum" in the original manuscript have been modified as "minimum and maximum" in the revised manuscript.

4.6 Add a blank between 1 and km. Check it in other places.
Response: The authors are sorry for these errors. The authors have examined the original manuscript and corrected them in the revised manuscript.

4.7 It is suggested to add one sentence to describe larger DISO and better result here. Just like it is mentioned in L276.
Response: The authors have added "Overall, the smaller DISO is, the better estimates are" after Equation (8) to qualitatively describe the implication of DISO in the revised manuscript.

4.8 L338 and L446, 2021a and 2018a, delete "a" for the paper from the same first author is only listed once. Please check others. L370, 375, 400, 405, is the first author right or the order of the first name and the second name should be changed? Please check others.
Response: (1) The authors are sorry for this. The authors have checked and corrected the problem of "L338 and L446, 2021a and 2018a". (2) They are not the first names of the authors in L370, 375, 400, 405.

**Reviewer #2:**
Qin et al. construct a 60–year (1961–2020) near-surface air temperature dataset over the glaciers of the Tibetan Plateau by fusing satellite and multi-source observations. The used ensemble learning model is described in detail and sufficient experiments are conducted to validate the reliability of constructed datasets. The manuscript is well organized, and all results are clearly presented.

Minor comments:

1) As the article is aimed at a data journal, I think it is better to include more key information on the dataset in the title of the manuscript, such as the spatial resolution (1km), temporal resolution (monthly) and used methods (data fusion or machine learning).

Response: The authors have changed the title to "A long-term 1-km monthly near-surface air temperature dataset over the Tibetan glaciers by fusion of station and satellite observations".

2) Section 3.2: It is necessary to present some reasons or your considerations for selecting the random forest model, by citing relevant literatures or adding concise discussions.

Response: As the reviewer points out, there are many machine learning methods that could be selected to convert LSTs to SATs. There has already existed a study which compare a total of ten machine learning methods (including several methods mentioned by the reviewer) in converting LSTs to SATs. The result shows that Cubist and random forest rank the first two places and their difference is subtle. Moreover, the random forest has been successfully applied in many studies to convert LSTs to SATs. Therefore, the authors have added description and citation "which has been proved effective in many scenarios (Belgiu and Dragut 2016; Xu et al. 2018)" in the revised manuscript.

3) Eq. 8: Please give more description for DISO. How should readers interpret the value? Does a larger or smaller value mean better?

Response: The authors have added "Overall, the smaller DISO is, the better estimates are" after Equation (8) to qualitatively describe the implication of DISO in the revised manuscript.

4) Please add legends for all scatterplots.

Response: The authors have modified all scatter plots in the revised manuscript by adding the color bars. These newly drawn figures are displayed in the following.

[Figure]

Figure 3.

[Figure]

Figure 4.

[Figure]

Figure 5.

Figure 6.

Figure 7.

5) Whether the stations outside the TP in the Northern Hemisphere locate at glaciers? If yes, it is better to add a sub-figure in Figure 4 to show the validation results at such station.

Response: These stations outside of the TP are not at glaciers. But we will try our best to collect records at glaciers, and we will consider more glacier stations all over the world the subsequent research.